# Monkeypox-Related Stigma and Vaccine Challenges as a Barrier to HIV Pre-Exposure Prophylaxis among Black Sexual Minority Men

**DOI:** 10.3390/ijerph20146324

**Published:** 2023-07-08

**Authors:** Rodman E. Turpin, CJ Mandell, Aaron D. Camp, Rochelle R. Davidson Mhonde, Typhanye V. Dyer, Kenneth H. Mayer, Hongjie Liu, Thomas Coates, Bradley O. Boekeloo

**Affiliations:** 1Department of Global and Community Health, College of Public Health, George Mason University, Fairfax, VA 22030, USA; cmandel3@gmu.edu (C.M.); rmhonde@gmu.edu (R.R.D.M.); 2Heller School for Social Policy and Management, Brandeis University, Waltham, MA 02453, USA; aaron.coleman@inova.org; 3INOVA Health System, Fairfax, VA 22031, USA; 4Department of Epidemiology and Biostatistics, School of Public Health, University of Maryland, College Park, MD 20742, USA; typhanye@umd.edu (T.V.D.); hliu1210@umd.edu (H.L.); 5The Fenway Institute, Fenway Health, Boston, MA 02215, USA; kmayer@fenwayhealth.org; 6Department of Medicine, Beth Israel Deaconess Medical Center, Boston, MA 02215, USA; 7Harvard Medical School, Harvard University, Boston, MA 01451, USA; 8David Geffen School of Medicine, University of California, Los Angeles, CA 90095, USA; tcoates@mednet.ucla.edu; 9Department of Behavioral and Community Health, School of Public Health, University of Maryland, College Park, MD 20742, USA; boekeloo@umd.edu

**Keywords:** HIV, PrEP, mpox, race, sexuality, intersectionality

## Abstract

Background: The U.S. monkeypox (mpox) outbreak of 2022 was a unique emergent public health crisis disproportionately affecting Black sexual minority men (BSMM). Similar to other stigmas, mpox-related stigma may have adverse effects on BSMM, including deterring HIV prevention such as PrEP. Methods: Our study investigated the experiences and perceptions of BSMM related to mpox, including mpox-associated stigma, and PrEP engagement among BSMM. We conducted qualitative interviews of 24 BSMM attending HIV prevention-related events in the greater D.C. Metropolitan area. In-depth interviews were conducted via phone, and responses to questions specific to the mpox outbreak were analyzed using thematic analysis. Results: We identified three key themes from the analysis: Mpox-related stigma, Mpox vaccine availability concerns, and Mpox vaccine hesitancy. Participants also described relationships between each of these three themes and PrEP use. Mpox stigma was particularly relevant as it is related to sexual stigma and is a deterrent to PrEP use. A sense of health system neglect of BSMM, especially related to low mpox vaccine availability, was also described. Conclusions: We identified mpox stigma and challenges related to mpox vaccination as key themes among BSMM, with implications for PrEP use. Future research exploring medical mistrust among BSMM, particularly related to HIV prevention, is recommended.

## 1. Introduction

Monkeypox, also referred to as “mpox” by the World Health Organization (WHO), is an uncommon disease caused by the same family of viruses as smallpox [1,2]. Before the 2022 outbreaks, mpox cases were recorded in humans within central and western African countries since 1970 [2]. Most infections have been recorded amongst men who have sex with men (MSM); however, any person can be infected through sexual and non-sexual exposures, including more general close contact. In May 2022, approximately 200 suspected cases were reported from North America, Europe, Australia, and Israel; in the same month, the CDC began an emergency response to investigate cases of mpox in nine states within the U.S. [3]. By the end of May, 17 cases were found, of which 16 were diagnosed in men who identified as gay, bisexual, or MSM. The disproportional impact amongst sexual minority men led to their categorization as a higher-risk group. Over 100 LGBTQ+ organizations advocated for available funding and supplies to be given to sexual health clinics and community-based organizations in order to combat the rise of mpox cases [4]. Notably, more than half of the mpox cases in the U.S. have been among Black and Hispanic sexual minority men (SMM) under the age of 40 [2].

The greater rates of mpox among SMM also lead to greater vulnerability to mpox-related stigma and discrimination, including harassment and violence from peers and society at large [5,6,7,8]. This is particularly true for Black sexual minority men (BSMM), who are disproportionately represented among mpox cases compared to SMM of other racial/ethnic groups [2]. Notably, infectious disease-related interpersonal stigma, particularly from peers, is often a barrier to public knowledge related to these diseases; we observe this in mpox-related stigma as well. A study by Ogunbajo et al. found that BSMM participants had high levels of anticipated mpox-related stigma, including some experiences with monkeypox stigma, and were notably skeptical about the seriousness of mpox [9]. Mpox-related discrimination can also include experiencing harassment and violence [10]. Notably, BSMM are often vulnerable to mpox-related stigma whether they have had the infection or not, as the disease is often incorrectly conflated with sexual orientation [7]. This may be particularly true for BSMM, given that this population is not only more vulnerable to mpox than SMM of other races but is often more marginalized based on the intersection of both sexual identity and race [11,12]. At this intersection, BSMM concurrently face interpersonal and structural racism and homophobia, including barriers to equitable healthcare, such as a lack of culturally competent care. This exacerbates medical distrust and further marginalizes this population, creating an even more heightened vulnerability to stigma, including mpox stigma.

Mpox stigma can lead to significant minority stress. Minority stress theory posits that people of marginalized identities experience additional stressors related to stigma and discrimination that create accumulated adverse health effects [13]. The adverse health effects often include negative mental health outcomes such as depression and anxiety, substance use, and suicidality [14,15,16,17]. In the context of BSMM and healthcare service utilization, a significant barrier to accessing healthcare is stigmatization from healthcare personnel or healthcare systems that have discriminatory practices [18,19,20,21]. Stigma has been a substantial barrier to the engagement of BSMM with HIV prevention services, including PrEP use [9,10,11,12,19,20,21]; this is likely to be true for mpox-related stigma as well [6,7,8,10]. Mpox-related stigma may deter PrEP use by discouraging engagement with HIV prevention organizations that often provide PrEP, and creating overall marginalization that socially isolates BSMM, cutting off important social network connections to community-based PrEP linkage. Community-based PrEP services are especially important for BSMM given that BSMM often trust these institutions more than larger healthcare organizations. How mpox-related stigma affects PrEP use is especially concerning for BSMM, as PrEP is one of the most effective tools in preventing new HIV infections. This is especially relevant for BSMM, who are substantially more vulnerable to HIV acquisition than their white or heterosexual peers [22]. For these reasons, it is important to understand how mpox stigma affects PrEP use among BSMM.

Our study investigated two research questions related to BSMM and mpox. Broadly, what were the social experiences and perceptions of BSMM related to the mpox outbreak? More specifically, how do these experiences and perceptions, including mpox-related stigma, affect PrEP acceptability and uptake among BSMM? Our study findings can help highlight the relevance of mpox-related stigma towards BSMM, and identify connections between mpox-related stigma, identity-based discrimination, and HIV prevention efforts for this community.

## 2. Materials and Methods

### 2.1. Recruitment and Sample

We approached potential participants at community-based events in the D.C. Metropolitan area (e.g., D.C., Maryland, Virginia), as this is an area with disproportionately elevated HIV prevalence [22], with several community-based programs specifically for BSMM. Events were focused on health, social connection, and overall wellness for BSMM. These included events held in healthcare organizations, bars, and other social venues focused on BSMM, as these are ideal for reaching this community. At the end of each event, we discussed the study’s overall goals with attendees and assessed their eligibility based on specific criteria, including being 18 years or older, male, Black or African-American, having had a same-sex partner in the past six months, and having attended a BSMM-specific health intervention event in the past year (e.g., a health education program primarily marketed to and attended by BSMM). Additionally, while not being part of the eligibility criteria, all participants self-reported a sexual minority identity and resided in the greater D.C. Metropolitan area. Attendees were encouraged to refer others who met the same criteria. Those who expressed interest and met eligibility requirements were scheduled for in-depth interviews and given an electronic consent form to sign ahead of time. This process happened within the following week.

### 2.2. Interview Procedures

We carried out detailed phone interviews where the interviewer thoroughly explained the consent form to participants, addressing any concerns they may have had and obtaining their informed consent before commencing each interview. A semi-structured interview guide was utilized during the process, with each interview lasting approximately 20 to 30 min. Questions for this study aim were primarily focused on three specific areas: general concerns related to mpox (e.g., “What are your thoughts and concerns related to the current mpox outbreak?”), relationships to PrEP engagement (e.g., “Has monkeypox affected your considerations of using PrEP”, “Has the monkeypox outbreak affected how you access and use PrEP?”), and general sociodemographics (e.g., “How old are you?”, “What is your sexual identity?”). These questions were selected to explore BSMM experiences and beliefs related to the mpox outbreak. All interviews were conducted by the study PI, who is a BSMM community member. We conducted 24 interviews, as we suspected this would be sufficient to achieve saturation of themes in our qualitative data analysis. All participants were compensated $30 for each interview.

### 2.3. Data Management

To ensure accuracy, all interviews were recorded and then transcribed in two steps. First, we utilized a transcription service called Descript, which transcribed audio into text that could be edited using word processing software [23]. Second, a graduate student carefully scrutinized both the transcripts and recordings, making corrections to any errors in the initial transcription. Note that the automated transcription was already approximately 90% to 95% accurate, with only minimal manual corrections needed. To store the transcript data, both Descript and Microsoft Word were utilized, and various precautions were taken to maintain data security and confidentiality [23]. For instance, audio data were analyzed only on encrypted and password-protected computers, disconnected from public networks. Audio data were not taken off-site and were deleted from Descript after transcription. Additionally, all identifiable information was removed from the transcripts. All study procedures were approved by the George Mason University and University of Maryland, College Park institutional review boards.

### 2.4. Thematic Analysis

The analysis of the interview data involved a team comprising two professors and two postgraduate students, two of whom were part of the BSMM community, including the primary investigator. To recognize and delineate the patterns in the data, an inductive approach was adopted, guided by the six stages of thematic analysis. Phase 1—becoming familiar with the data: one member of the analysis team initiated the process by reading and re-analyzing each transcript independently, making note of important topics and inquiries. Phase 2—generating initial codes: Next, the researchers pinpointed specific sections of the interviews and noted recurring ideas. They met biweekly to discuss their findings. Phase 3—searching for themes: The analysis team met biweekly to review and discuss passages and codes that were identified. Then one researcher acted as the main coder and analyzed the interviews for common themes, using these themes to categorize relevant sections and categorizing text passages based on the themes identified. This was followed by a secondary coder allowing for the performance of an interrater reliability check, assessing agreement between the two assessments and all primary codes. A second coder then reviewed the work to ensure accuracy and also added any missed codes. Any discrepancies were discussed and resolved at meetings between both coders. Phase 4—interpreting the themes: once the coding process was complete, the main coder identified keywords and phrases that summarized each theme. Phase 5—refining the specifics of the themes: the team then reviewed these words and grouped similar ones together as either main themes or sub-themes. Phase 6—final analysis: the final list of themes was interpreted by the analysis team to identify both themes related to thoughts on mpox overall, as well as specifically how mpox may affect PrEP acceptability, access, and utilization.

## 3. Results

### 3.1. Sample Description

The sample consisted of 24 Black sexual minority men (Table 1). Three-quarters of the sample fell between the ages of 25 to 44 (the full range was 18 to 49). A little over half of the individuals’ highest levels of education were college, while for a quarter it was a high school diploma. Almost two-thirds lived in Maryland, with the remainder located in Virginia or Washington D.C. Among the group, two-thirds had never taken PrEP, while just one-fifth of the participants were presently using it. The majority of the sample (72.2%) had concerns related to anticipated mpox stigma (e.g., worries about experiencing mpox-related stigma), and individuals with greater anticipated mpox stigma were more likely to not consider PrEP use. Additionally, individuals living in Maryland had greater proportions of high anticipated mpox stigma than those living in D.C. or Virginia. We identified three key themes from the transcript analysis process: mpox-related stigma, mpox vaccine availability concerns, and Mpox vaccine hesitancy (Table 2). Participants also described relationships between each of these three themes and PrEP use. Pseudonyms for all participants are provided in quotations.

### 3.2. Mpox-Related Stigma

By far, the single most common concern related to mpox was anticipating stigma due to having or being perceived to have mpox (91.6%). Note that this includes both personally experienced and anticipated mpox stigma. Mpox stigma was noted as particularly related to sexual stigma, with the idea that being perceived to have mpox would lead to people making negative assumptions about the person’s sexual activity. This is described by one participant, “Jared”, in reference to both the media and peers:

“*I think on several fundamental levels, monkeypox is much more terrifying from the fact that like it was stigmatized and directed towards our community… the narrative that I was exposed to was that there was a lot more like, unprotected sex that helped facilitate the spread of monkeypox. It’s just like, so insidious. I don’t want people to think that about me.*”

Many participants mentioned how mpox was assumed to be a “gay disease”. “Keenan” described this stigma as a channel for the more general homophobic stigma, particularly from heterosexual peers and family members, that accompanies HIV-related stigma:

“*I’ll say, you know, the scary thing about it is the fact that, you know, again, it’s been deemed as the gay disease and that’s not necessarily true. It definitely mirrored and echoed all the stigmas and homophobia that we see historically with HIV in the past. And um, with that, anything that mirrored HIV is also gonna have an effect on PrEP.*”

The parallels to the early AIDS epidemic were noted in several ways. In addition to mpox and HIV both being seen as a “gay disease”, both were described as largely neglected by the government, particularly related to the lack of mpox vaccine availability, a theme discussed later in our study. Sexual stigma was also described as a driver of both mpox and HIV stigma, with the idea that both conditions were falsely treated as an indicator of “high-risk” sexual behavior.

Notably, with rare exceptions, the actual symptoms of mpox were only raised as a concern related to visibility. The symptoms were primarily mentioned as an issue not because of the pain or any forms of physical debilitation, but because visible sores would make it apparent that the individual had mpox. This was reinforced by numerous participants, including “Keenan” and “Antoine”:

“*I think the recent rise probably has scared people because I think with Monkeypox, more people were worried about their vanity….with monkeypox, it is visible, it’s sores and stuff like that.*”

“*Well, Monkeypox has been a thing of its own, you know. A lot of people were against getting the vaccine, but once they found out, you know what Monkeypox looks like, visibly on someone, a lot of people ran out and got that.*”

It should be noted that while there were general appearance concerns, much of the fears around visibility were related to becoming vulnerable to stigma if others could visually identify that the individual may have mpox. This stigma was also described as related to the general lack of knowledge around mpox, as it was an emergent infectious disease. Multiple participants described how stigma related to mpox was directly a barrier to using PrEP services. “Marques” described this the most clearly, with sexual stigma related to mpox being a direct deterrent to engaging with PrEP service providers, with even the appearance of seeking care for mpox having significant sexual stigma attached to it (e.g., going to a BSMM-focused health care organization may suggest one is seeking care for mpox, even if they are only there for HIV prevention services):

“*No sir, I cannot go there (the BSMM clinic for PrEP) right now. Not until this (the mpox outbreak) calms down. I can pick the PrEP back up later, but if I step into that clinic and people even think I have monkeypox it’s done. They’ll say I got it at a hookup, or like a sex party....Everybody talks. And what if it gets back to someone I’m talking to (dating)? All of a sudden I’m fuckin’ around. Not worth it.*”

### 3.3. Mpox Vaccine Availability

Another common concern identified (70.8%) was related to the lack of available mpox vaccines. Participants frequently expressed frustration with the government’s response to the mpox outbreak. They mentioned the overall lack of availability of the vaccines, as well as the screening questionnaires used to ration vaccines, which were seen as personally invasive due to the sexual questions. Some participants even described seeking vaccines overseas due to the lack of vaccine availability in the U.S., such as “Jared”:

“*So, yeah, that definitely like shut my shit down, for a hot minute until I went to Canada. Yeah. I went to Canada cause Canada cares about its people. And made vaccines available to anyone and everyone who was around. And I am very thankful for it.*”

Another issue that some participants mentioned was changing guidelines around dosing, particularly near the end of 2022, when delivering one-fifth of a vaccine dose as a way to ration them more effectively was implemented. This caused concerns with many members of the BSMM community broadly, due to the uncertainty that a partial vaccine would have comparable efficacy to a full vaccine dose. “Amari” mentions these concerns as well:

“*And of course, our community mobilized to help get things taken care of. Suddenly there was an issue with, there weren’t enough vaccinations. And then, you know, changing the way in which the, the dosage was applied.*”

The “community mobilization” here reflects the role that BSMM community-based organizations have had in increasing vaccine availability for this community. Participants described that BSMM community-based organizations quickly became a critical venue of access for BSMM who otherwise would not have been able to acquire them, either due to transportation access challenges, socioeconomic barriers, or an overall lack of available vaccines. Finally, some participants mentioned the lack of vaccine accessibility as limiting opportunities for PrEP use. Given that SMM are priority populations for both mpox vaccination and PrEP, mpox vaccination could be a significant opportunity to promote PrEP use as well. This could allow for an effective integrated approach to SMM sexual health. “Amari” also describes how this could be implemented:

“*I would say, you know, that Monkeypox would have served or could have served as a precursor to the idea of really taking care of our sexual health, our sexual realness, and being in a space to get the vaccine. Hopefully care providers were pushing and suggesting, “hey, like there is this situation, you’re in for the vaccination, but have you thought more about PrEP?*”

### 3.4. M-Pox Vaccine Resistance

Finally, vaccine resistance was the third most common theme (50.0%). Many participants described hesitancy around biomedical prevention broadly as a longstanding issue among Black communities, primarily rooted in medical abuses and exploitation of Black people. This applies to mpox vaccination as well. Even with the reported issues related to vaccine availability, these fears are still omnipresent. “Deandre” described this when asked about the mpox outbreak:

“*We have this history of medical persons, medical ethics that go negatively and use our bodies for scientific research that has been unfortunate for our communities. But I am seeing in my space, in my conversations with people, a bit of hesitancy around any type of medication. Including with the vaccine and how Monkeypox took shape. And it was quick.*”

Multiple participants expressed that mpox vaccines can have some of the same sexual stigmas as PrEP use, particularly in people being perceived as engaging in sexually risky behaviors if they take the mpox vaccine (e.g., “behaving like they’re invincible”). This is compared to the stigmatizing idea that people who take PrEP are all sexually risky (e.g., “Truvada whores”). The similarities between mpox vaccine resistance and PrEP resistance were remarkable, and consistently described. One participant, “Marcell”, mentions this in detail:

“*One of the common things I saw people say or reference, kind of in an insulting way, is PrEP and monkeypox vaccines contribute to people feeling or behaving like they’re invincible. Like, I see people say in response to PrEP, well, “PrEP doesn’t protect you against all STIs”. And they’re using that as a, as a justification for not using PrEP at all. And in the same way with Monkeypox, as a justification for not getting a vaccine.*”

Related to PrEP use, a unique link between mpox vaccine resistance and PrEP hesitancy was exasperation with cumulative biomedical prevention needs relating to BSMM having sex. The constantly growing need to have various new biomedical interventions for sex can not only be somewhat exhausting, but it can also create a sense of futility that the current forms of prevention are never enough, and there will always be some emerging threat to sex for BSMM. “Jamal” described observing this phenomenon:

“*I’m seeing some people become more hesitant as first PrEP, then the COVID vaccine, then the Monkeypox vaccine. And it becomes like this endless list of, of medical interventions that you need to be able to have sex and that can be overwhelming. I see some people just kind of exhausted. And just like, oh, well it doesn’t matter. Just every, everything’s gonna get you. If it’s not COVID, it’s HIV. It’s not HIV, it’s Monkeypox. Like some people just get overwhelmed and say skip it all… And in terms of how it impacts PrEP, I think people are beginning to feel a little turned off again about the idea of using medicine and medicating themselves for situations.*”

## 4. Discussion

We found that fear of being stigmatized while seeking the mpox vaccine and challenges related to obtaining mpox vaccination were key themes among BSMM in our sample, and both had implications for PrEP use. The relationships between fear of being stigmatized for seeking the mpox vaccine as well as being stigmatized for seeking PrEP and hesitancy for obtaining both the mpox vaccine and PrEP [21,24,25,26,27]. Fear of being stigmatized was overwhelmingly the primary concern participants had regarding the mpox vaccine. This included fear of being labeled as having risky sexual behavior, infidelity related to partners, or generally being viewed as undesirable. This fear of stigma from seeking the mpox vaccine was so profound that it affected how participants engage with PrEP, with one participant even avoiding his usual PrEP clinic for fear of being assumed to have mpox. This highlights the profound relevance of stigma to not only PrEP promotion efforts but BSMM health efforts overall. The discussion of mpox stigma being related to sexual stigma is also especially salient considering sexual stigma was described as a deterrent to both PrEP use and mpox vaccination. This highlights the importance of addressing this stigma as a form of overall SMM sexual health. Given that BSMM have a disproportionately greater need for HIV prevention efforts, our findings here are especially relevant for public health strategies to combat the HIV epidemic [22]. While mpox is not as strong a public health priority as it was in the year prior, our study findings still highlight how infectious disease outbreaks, particularly those disproportionately among SMM communities, can create challenges that have implications for HIV prevention efforts in these communities. This is especially true for stigma related to infectious diseases.

Related to mpox vaccination challenges, we found two seemingly opposed difficulties: A lack of available mpox vaccines for BSMM, and vaccine hesitancy among BSMM. These affect PrEP use in different ways. Participants noted the failures in mpox vaccine availability as missed opportunities for PrEP outreach. This highlights the utility of integrated SMM health outreach efforts. An appointment for a mpox vaccine can be a venue to deliver PrEP messaging, as well as many other health services for BSMM [28]. This approach may be effective in future research and policy efforts to provide PrEP use to this population. Additionally, a lack of available mpox vaccines further contributes to a sense of health system neglect of BSMM, akin to the lack of government prioritization of HIV among SMM in the early AIDS epidemic. Multiple participants noted that such neglect is another example of the parallels between mpox and HIV [29]. Many participants in our sample had greater concerns about vaccine availability, which may lead to more fears related to contracting mpox due to the absence of vaccine protection. These fears about vaccine availability are critical given the aforementioned relationships between mpox stigma and PrEP use. This sense of neglect may further erode trust in health systems, facilitating distrust of other health services geared towards BSMM, including PrEP providers.

The findings related to participants reporting that mpox vaccines have many of the same sexual stigmas as PrEP use, particularly in people being perceived as engaging in sexually risky behaviors if they take the mpox vaccine [30]. Additionally, participants described exhaustion with the ever-growing biomedical interventions for BSMM sex. This exhaustion may reflect a sense of sex among BSMM being pathologized, which results in BSMM being pathologized. It is critical to ensure that while prevention efforts effectively address disproportionate vulnerability to adverse health outcomes, they do not stigmatize or pathologize the communities they aim to serve. In many ways, our findings highlight ubiquitous health equity issues that exceed mpox alone and have implications for overall prevention efforts for BSMM [29].

This study has important limitations. Our findings may not apply to BSMM populations in other regions or to SMM of different races and ethnicities, though its emphasis on BSMM is justified by their heightened vulnerability to both mpox and HIV acquisition. Additionally, the D.C. Metropolitan area’s high HIV prevalence makes it a crucial area for understanding PrEP use. Although our sample size was small, we were able to gain a thorough understanding of participant experiences due to the saturation of themes. Although social desirability bias may have influenced responses on sensitive topics such as mpox stigma, the rapport established with the interviewer enabled participants to share their vulnerable and stigmatized experiences.

## 5. Conclusions

Overall, we identified mpox stigma and challenges related to mpox vaccination as key themes among BSMM in our sample, with substantial implications for PrEP use. While focused on mpox, our findings highlight several areas relevant to PrEP promotion efforts among BSMM more broadly. This study emphasizes the relevance of destigmatization and providing priority health services to BSMM communities to directly address health equity efforts and foster greater medical trust. The importance of not stigmatizing or pathologizing marginalized communities in health messaging, including BSMM communities, cannot be overstated. Future areas of research exploring the complexities between health messaging and medical trust among BSMM, particularly related to HIV prevention, are recommended.

## Figures and Tables

**Table 1 ijerph-20-06324-t001:** Anticipated mpox stigma and stratified characteristics of interviewed Black sexual minority men (*n* = 24).

	Total (*n* = 24)	No Anticipated Mpox Stigma (*n* = 6)	Anticipated Mpox Stigma (*n* = 18)
Age			
18–24	20.8% (5)	**0.0% (0)**	**27.8% (5)**
25–34	37.5% (9)	**66.7% (4)**	**27.8% (5)**
35–44	33.3% (8)	33.3% (2)	38.9% (7)
45–49	8.3% (2)	0.0% (0)	11.1% (2)
Ethnicity			
Non-Hispanic/Latino	83.3% (20)	83.3% (5)	83.3% (15)
Hispanic/Latino	16.7% (4)	16.7% (1)	16.7% (3)
Highest Education Level			
High School	25.0% (6)	16.7% (1)	**27.8% (5)**
Undergraduate College	58.3% (14)	66.7% (4)	55.6% (10)
Graduate College	16.7% (4)	16.7% (1)	16.7% (3)
State of Residence			
District of Columbia	20.8% (5)	**33.3% (2)**	**16.7% (3)**
Maryland	62.5% (15)	**33.3% (2)**	**72.2% (13)**
Virginia	16.7% (4)	**33.3% (2)**	**11.1% (2)**
PrEP Use			
Never	66.7% (16)	**50.0% (3)**	**72.2% (13)**
Previous Use	12.5% (3)	16.7% (1)	11.1% (2)
Current Use	20.8% (5)	**33.3% (2)**	**16.7% (3)**

Parenthesized values in cells are absolute frequencies. Proportion differences >15% are bolded.

**Table 2 ijerph-20-06324-t002:** Mpox-related themes identified in interviews with Black sexual minority men (*n* = 24).

Mpox stigma (91.6%)	Mpox sexual stigma (91.6%), Vulnerability to mpox stigma due to visibility (75.0%), Mpox stigma similarities to HIV stigma (66.7%), Mpox-related homophobia (29.2%), Mpox stigma deterring PrEP (54.1%)
Mpox vaccine availability (70.8%)	Needs for more mpox vaccines (70.8%), governments not prioritizing mpox vaccines (66.7%), Mpox vaccine unavailability facilitating medical mistrust (37.5%), concerns about reduced vaccine dose for rationing (20.8%), Lost opportunities for PrEP promotion alongside mpox vaccines (20.8%)
Mpox vaccine hesitancy (50.0%)	Mpox vaccine hesitancy (50.0%), mpox vaccine-related sexual stigma (37.5%), exhaustion with medical interventions for SMM sex (16.7%)

Parentheses after each theme contain the percentage of participant interviews where that theme was observed.

## Data Availability

Data is available upon request. Please contact Rodman Turpin (Rturpin@gmu.edu) regarding data requests.

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
