# Peer review of "Monkeypox-Related Stigma and Vaccine Challenges as a Barrier to HIV Pre-Exposure Prophylaxis among Black Sexual Minority Men"

_ijerph, 2023, doi:10.3390/ijerph20146324_

Round 1

Reviewer 1 Report

This paper presents some very interesting points: the sense of neglect and stigma (gay-related diseases) of Mpox (similar to HIV historically); as well as uptake of Mpox vaccines and PrEP use (stigma related to male-to-male sex, social disadvantages and risk of sexual transmission).

The authors lightly touched upon minority stress theory in the intro section. To further highlight these multi-layered stigma (perceived or enacted) across the impact of Mpox and uptake of Mpox vaccinations, I would suggest a bit more discussion in terms of intersecting stigmas among these black MSM (race, same-sex partnership, access and trust to biomedical interventions) in light of equitable health care access and engagement/empowerment (through community consultation, mobilisation) with gender and sexual minority populations.

Author Response

Reviewer 1

  1. This paper presents some very interesting points: the sense of neglect and stigma (gay-related diseases) of Mpox (similar to HIV historically); as well as uptake of Mpox vaccines and PrEP use (stigma related to male-to-male sex, social disadvantages and risk of sexual transmission).

We thank the reviewer for the positive assessment.

  1. The authors lightly touched upon minority stress theory in the intro section. To further highlight these multi-layered stigma (perceived or enacted) across the impact of Mpox and uptake of Mpox vaccinations, I would suggest a bit more discussion in terms of intersecting stigmas among these black MSM (race, same-sex partnership, access and trust to biomedical interventions) in light of equitable health care access and engagement/empowerment (through community consultation, mobilisation) with gender and sexual minority populations.

We have revised to clarify this more (1. Introduction)

“BSMM are often vulnerable to mpox-related stigma whether they have had the infection or not, as the disease is often incorrectly conflated with sexual orientation [7]. This may be particularly true for BSMM, given that this population is not only more vulnerable to mpox than SMM of other races, but is often more marginalized based on intersections of both sexual identity and race [11,12]. At this intersection, BSMM concurrently face interpersonal and structural racism and homophobia, including barriers to equitable healthcare, such as a lack of culturally competent care. This exacerbates medical distrust and further marginalizes this population, creating an even more heightened vulnerability to stigma, including mpox stigma.”

“Mpox-related stigma may deter PrEP use through discouraging engagement with HIV prevention organizations that often provide PrEP, and creating overall marginalization that socially isolates BSMM, cutting off important social network connections to community-based PrEP linkage. Community-based PrEP services are especially important for BSMM given that BSMM often trust these institutions more than larger healthcare organizations.”

Reviewer 2 Report

The authors' contribution is significant, largely innovative, and helpful in understanding the challenges experienced by BSMM as they related to key aspects of their health and health care.    

My key concern is that the presentation of methods is a bit thin. Specifically, why DC Metro area?  Why community events and what kind of community events?  What was the sampling strategy at these events?  Why 24? 

In the reporting of results, Black or African American is not listed although is stated in the Methods.

Author Response

Reviewer 2

  1. The authors' contribution is significant, largely innovative, and helpful in understanding the challenges experienced by BSMM as they related to key aspects of their health and health care.    

We thank the reviewer for the positive assessment.

  1. My key concern is that the presentation of methods is a bit thin. Specifically, why DC Metro area?

We revised to clarify this (2.1 Recruitment and Sample):

We approached potential participants at community-based events in the D.C. Metropolitan area (e.g., D.C., Maryland, Virginia), as this is an area with disproportionately elevated HIV prevalence [22] and several community-based programs specifically for BSMM.

  1. Why community events and what kind of community events? 

We revised to more clearly describe these events and our reasoning (2.1 Recruitment and Sample):

Events were focused on health, social connection, and overall wellness for BSMM. These included events held in healthcare organizations, bars, and other social venues focused on BSMM, as these are ideal for reaching this community.

  1. What was the sampling strategy at these events?  Why 24? 

We have revised to clarify this (2.2 Interview Procedures):

“We conducted 24 interviews, as we suspected this would be sufficient to achieve saturation of themes in our qualitative data analysis.”

We also revised to more clearly describe our sampling approach at events (2.1 Recruitment and Sample):

We approached potential participants at community-based events in the D.C. Metropolitan area (e.g., D.C., Maryland, Virginia), as this is an area with disproportionately elevated HIV prevalence [22], with several community-based programs specifically for BSMM. Events were focused on health, social connection, and overall wellness for BSMM. These included events held in healthcare organizations, bars, and other social venues focused on BSMM, as these are ideal for reaching this community. At the end of each event, we discussed the study's overall goals with attendees and assessed their eligibility based on specific criteria, including being 18 years or older, male, Black or African-American, having had a same-sex partner in the past six months, and having attended a BSMM-specific health intervention event in the past year (e.g., a health education program primarily marketed to and attended by BSMM).”

  1. In the reporting of results, Black or African American is not listed although is stated in the Methods.

We have revised to clarify this (3. Results):

“The sample consisted of 24 Black sexual minority men (Table 1).”

Reviewer 3 Report

Initially, I consider it important to congratulate the group for the work presented, it is also relevant to say that the writing is well prepared and that some points can be better portrayed/improved considering the reader's perspective.

Section - Methods

I believe that the description of topic 2.4 would be clearer if right after the first sentence ending in - guided by the six stages of the thematic analysis - the phase/stage were presented first and then its description, in a kind of list

Phase 1 - becoming familiar with the data: description

Phase 2 - generating initial codes: description

And so on until you reach the final stage.

And in each of these phases it could be indicated who the coder is (PI or student?), so it would be clearer and better explained who did what, this would reduce doubts in excerpts like this... " the researchers pinpointed specific sections of the interviews and notes recurring ideas (lines 143 and 144) ... who are the researchers? The team formed by 2 professors and two students? Or could I consider that the researchers are in fact the same analysis team? If so, it would be better to use the denomination already seen in line 140 (analysis team).

Table 1 should also present the absolute values and not just the relative ones (percentages)

Author Response

Reviewer 3

  1. Initially, I consider it important to congratulate the group for the work presented. It is also relevant to say that the writing is well prepared and that some points can be better portrayed/improved considering the reader's perspective.

We thank the reviewer for the positive assessment.

  1. I believe that the description of topic 2.4 would be clearer if right after the first sentence ending in - guided by the six stages of the thematic analysis - the phase/stage were presented first and then its description, in a kind of list: Phase 1 - becoming familiar with the data: description, Phase 2 - generating initial codes: description, And so on until you reach the final stage.

We have revised this section to match this format (2.4 Thematic Analysis):

“To recognize and delineate the patterns in the data, an inductive approach was adopted, guided by the six stages of thematic analysis. Phase 1 – becoming familiar with the data: One member of the analysis team initiated the process by reading and re-analyzing each transcript independently, making note of important topics and inquiries. Phase 2 - generating initial codes: Next, the researchers pinpointed specific sections of the interviews and noted recurring ideas. They met biweekly to discuss their findings. Phase 3 – searching for themes: The analysis team met biweekly to review and discuss passages and codes that were identified. Then one researcher acted as the main coder and analyzed the interviews for common themes, using these themes to categorize relevant sections and categorizing text passages based on themes identified. This was followed by a secondary coder allowing for performance of an interrater reliability check, assessing agreement between the two assessments and all primary codes. A second coder then reviewed the work to ensure accuracy and also added any missed codes. Any discrepancies were discussed and resolved at meetings between both coders. Phase 4 – interpreting the themes: Once the coding process was complete, the main coder identified keywords and phrases that summarized each theme. Phase 5 – refining the specifics of the themes: The team then reviewed these words and grouped similar ones together as either main themes or sub-themes. Phase 6 – final analysis: The final list of themes was interpreted by the analysis team  to identify both themes related to thoughts on mpox overall, as well as specifically how mpox may affect PrEP acceptability, access, and utilization.”

  1. And in each of these phases it could be indicated who the coder is (PI or student?), so it would be clearer and better explained who did what, this would reduce doubts in excerpts like this... " the researchers pinpointed specific sections of the interviews and notes recurring ideas (lines 143 and 144) ... who are the researchers? The team formed by 2 professors and two students? Or could I consider that the researchers are in fact the same analysis team? If so, it would be better to use the denomination already seen in line 140 (analysis team).

We have revised to clarify this was the same analysis team:

“Phase 6 – final analysis: The final list of themes was interpreted by the analysis team to identify both themes related to thoughts on mpox overall, as well as specifically how mpox may affect PrEP acceptability, access, and utilization.”

  1. Table 1 should also present the absolute values and not just the relative ones (percentages)

We have added the absolute frequencies for table 1, with an appropriate footnote.